# Early Covert Appearance of Marginal Zone B Cells in Salivary Glands of Sjögren′s Syndrome-Susceptible Mice: Initiators of Subsequent Overt Clinical Disease

**DOI:** 10.3390/ijms22041919

**Published:** 2021-02-15

**Authors:** Ammon B. Peck, Cuong Q. Nguyen, Julian Ambrus

**Affiliations:** 1Department of Infectious Diseases and Immunology, College of Veterinary Medicine, University of Florida, Gainesville, FL 32610, USA; nguyenc@ufl.edu; 2Division of Allergy, Immunology and Rheumatology, SUNY Buffalo School of Medicine, Buffalo, NY 14023, USA; jambrus@buffalo.edu

**Keywords:** Sjögren’s syndrome, C57BL/6.NOD-*Aec1Aec2* mice, marginal zones (MZ), marginal zone B (MZB) cells, Notch2, B cell receptor (BCR), sphingosine-1-phosphate (S1P), sphingosine-1-phosphate receptor (S1PR), RNA transcriptome microarrays

## Abstract

The C57BL/6.NOD-*Aec1Aec2* mouse model has been extensively studied to define the underlying cellular and molecular bioprocesses critical in the onset of primary Sjögren’s Syndrome (pSS), a human systemic autoimmune disease characterized clinically as the loss of lacrimal and salivary gland functions leading to dry eye and dry mouth pathologies. This mouse model, together with several gene knockout mouse models of SS, has indicated that B lymphocytes, especially marginal zone B (MZB) cells, are necessary for development and onset of clinical manifestations despite the fact that destruction of the lacrimal and salivary gland cells involves a classical T cell-mediated autoimmune response. Because migrations and functions of MZB cells are difficult to study in vivo, we have carried out ex vivo investigations that use temporal global RNA transcriptomic analyses to profile autoimmunity as it develops within the salivary glands of C57BL/6.NOD-*Aec1Aec2* mice. Temporal profiles indicate the appearance of Notch2-positive cells within the salivary glands of these SS-susceptible mice concomitant with the early-phase appearance of lymphocytic foci (LF). Data presented here identify cellular bioprocesses occurring during early immune cell migrations into the salivary glands and suggest MZB cells are recruited to the exocrine glands by the upregulated Cxcl13 chemokine where they recognize complement (C’)-decorated antigens via their sphingosine-1-phosphate (S1P) and B cell (BC) receptors. Based on known MZB cell behavior and mobility, we propose that MZB cells activated in the salivary glands migrate to splenic follicular zones to present antigens to follicular macrophages and dendritic cells that, in turn, promote a subsequent systemic cell-mediated and autoantibody-mediated autoimmune T cell response that targets exocrine gland cells and functions. Overall, this study uses the power of transcriptomic analyses to provide greater insight into several molecular events defining cellular bioprocesses underlying SS that can be modelled and more thoroughly studied at the cellular level.

## 1. Introduction

B cells and B cell products, including auto-antibodies, have long been known to be important components of autoimmune responses in rheumatoid diseases (reviewed in [1,2,3]), especially systemic lupus erythematosus (SLE) [4,5] and Sjögren’s syndrome (SS) [6,7,8,9]. More recently, however, an increasing number of published reports indicate that marginal zones (MZ) and their associated marginal zone B cell populations per se interact to establish the correct underlying environment for both development and onset of the observed clinical pathologies. On the other hand, whether MZB cells are sufficient to cause rheumatoid autoimmunity remains an open question, since multiple immune cell types (i.e., T and follicular B cells, NK cells, macrophages and/or dendritic cells) have been shown to be present in the targeted tissues at the time of clinical disease onset and progression [10,11]. In addition, compositions of these different immune cells may be present or absent in the lesions at different stages of disease when examined by histology and FACS analyses, a fact that predicates the need for temporal studies as opposed to single timepoint analyses. This is clearly the case for SS, a systemic autoimmune disease characterized by an immune attack primarily against the salivary and lacrimal glands and having a skewed prevalence towards women presenting in clinics with a wide variety of symptoms [11]. Similar findings are strongly supported by studies in appropriate murine models [12,13,14,15,16].

More than two decades ago, studies by Robinson et al. [17] using the NOD.B10-*H2^b^*-*Igµ^−/−^* gene knockout (KO) mouse model revealed an absolute requirement for B cells in the development and onset of SS-like diseases that normally develops spontaneously in the parental NOD.B10-*H2^b^* autoimmune mice. Recent studies in both Baff [18,19,20] and B6.*Il14α* transgenic (TG) mice [15,21] not only support this earlier finding, but have shown that elimination of the MZB cell population(s) or blocking the lymphotoxin activity required for MZB cell ontogeny [22] prevents development of SS-like diseases, including lymphomagenesis.

MZB cells are bone marrow-derived B lymphocytes characterized by limited expression of immunoglobulin variable region genes encoding predominantly IgM antibodies of which many are auto-reactive [23]. In addition, they can function as innate and/or transitional cells capable of rapid responses to both T cell-independent and T cell-dependent antigens, as well as regulate activation of adaptive immune responses as antigen-presenting cells. Although MZB cells are strong responders to blood-borne pathogens, they also express B cell and ligand receptors capable of initiating diverse anti-self, autoimmune responses. Importantly, MZB cells have been identified within the salivary and lacrimal glands of both patients and mouse models with salivary gland disease [6,24,25].

While considerable information is now available concerning MZB cell features, how this cell population functions at the molecular level to promote exocrine gland disease remains unknown. For nearly two decades, we have utilized temporal global transcription profiling to establish interactive gene sets that define activated and/or non-activated cellular functions present in the salivary and lacrimal glands of mice undergoing development and onset of an SS-like pathology [25,26,27,28,29,30,31,32,33,34,35,36]. These studies have provided further evidence that MZB cells are recruited to the exocrine glands, apparently establishing the environment conducive for a subsequent destructive autoimmune attack. However, many regulated interactive bioprocesses of the disease activity remain to be defined. Here, we present a transcriptomic analysis focused on known MZB cell markers that are capable of further defining the underlying molecular profile of autoimmunity and activate bioprocesses in the salivary gland.

## 2. Results

### 2.1. The Lymphocyte and Autoimmune Profiles of Salivary Gland Lymphocytic Infiltration (LF) during SS Development

#### 2.1.1. Lymphocytic Foci (LF)

Based on histological and pathological analyses of the development and onset of SS-like diseases in our NOD/ShiLtJ and C57BL/6.NOD-*Aec1Aec2* mouse models, we were able to divide disease progression into three distinct phases [16]. In the early covert phase, an intense inflammation occurs, associated with increased glandular cell apoptosis marked by concomitant physiological, biochemical and cellular dysfunctions within the exocrine glands. In the second intermediate phase, this inflammatory response subsides, but is quickly followed by the third phase characterized by short-lived innate cellular immunity that transitions to a prolonged classical T cell-mediated autoimmune response. This overt clinical phase can be quantitated by glandular dysfunction whose onset corresponds with the appearance of lymphocytic foci (LF) formed by a wave of B cells recruited to the exocrine glands but subsequently overtaken by increasing numbers of T lymphocytes. Examples of an intermediate-phase LF for both the C57BL/6J and C57BL/6.NOD-*Aec1Aec2* mouse models are shown in Figure 1. While LF (or lymphocyte infiltrations) can be present in C57BL/6J mice, there is far less LF development and apparently less pathology. With recent evidence indicating that the early B cell infiltrates are largely MZB cells, we hypothesize that the early infiltrating MZB cells establish an extra-nodal germinal center-like environment for an ensuing classical T cell-mediated immune response that overtakes the MZB cell population(s) within the exocrine glands. The following studies were meant to examine this hypothesis.

#### 2.1.2. Early Temporal Transcriptome Expressions in Salivary Glands of SS^S^ Mice Identify Important Bioprocesses That Define the Pro-SS Autoimmune Response

In agreement with our hypothesis, together with salivary gland pathology and histology, our temporal global transcriptome profiling analyses of C57BL/6.NOD-*Aec1Aec2* salivary glands indicate an early activation of Notch2 and the Notch2 signal transduction pathway, as presented in Figure 2. Based on the specific upregulated genes concomitantly expressed, the signal transduction pathway defined is *Notch2 > Furin > Rfng/Mfng > Dll2 > Adam10 > γSecretase > Dtx1 > HDAC > Rbpk > Cbfat2 > Hes6*, a pathway leading to Notch-specific gene transcription capable of regulating multiple cell functions, including activations of Nfkβ1 (encoded by nuclear factor kappa b subunit 1) and type 1 interferon (Ifn1, encoded by *Ifna1*). This early appearance also corresponds with the first observation of migration and infiltration of B cells into the salivary glands noted histologically [32]. In addition, the transcriptome data reveal that MZB cells entering the salivary glands at 8 weeks of age undergo both ontogenesis and cell activation. Furthermore, by 16 weeks of age, the transcriptome profile reveals the second wave of Notch2 activation, thereby prolonging the activated state. While these profiles correspond with the onset of the inflammatory/innate response and initiation of the early adaptive response, respectively, within the salivary gland, they do not specify if this is a single continuous event or a time-dependent multiple independent action.

These data raise an important question: do these temporal transcriptome analyses of salivary glands further identify important pathological bioprocesses related to early MZB cell functions? In this regard and of major significance is the upregulated expression of *Tbk1*, the second marker associated with MZB cells. It is now recognized that Tbk1 is targeted by Dtx4 for degradation, a process that downregulates Irf3 and therefore synthesis of type 1 interferon (Inf1); thus, it is predicted that the observed downregulated expression of *Dtx4* would be accompanied by sustained upregulation of the *Tmem173 > Tbk1 > Irf3* signaling pathway regulating IFN synthesis. As shown in Figure 3, these three genes are strongly upregulated during disease development and most likely give rise to the extensive and well-defined IFN signature that is characteristic of SS.

#### 2.1.3. MZB Cells Are Multifunctional and Therefore Express a Broad Array of Receptors That Regulate Specific Activities

In addition to B cell receptors (BCRs), toll-like receptors (TLRs) and Notch2, MZB cells also express Cr2 (the complement receptor encoded by *Cd21*), S1pr (the sphingosine-1-phosphate receptor encoded by *S1pr4*), TACI (encoded by *Tnfrsf13b*) and IL22 receptor (encoded by *Il22ra*) [24,37]. Furthermore, MZB cells can be activated by MZ macrophages expressing DCIR (encoded by *Clec4a4*), macrophage receptor with collagenous structure (encoded by *Marco*) and SIGN-R1 (encoded by *Cd209a*) or by metallophilic macrophages expressing Cd169 (encoded by *Siglec1*) [38]. As shown in Figure 4, each of these genes, with the exception of *Siglec1*, exhibits upregulated expression in the salivary glands of C57BL/6.NOD-*Aec1Aec2* mice and does so in a simultaneous manner around 4 months of age. In contrast, no significant upregulated gene expression was observed for this same set of receptors in the salivary glands of C57BL/6J mice at the same age. In addition, genes encoding Tnfsf13 (April) and Tnfsf13b (Baff), the two specific ligands for Tnfrsf13b, also exhibit upregulated expression in the SS-susceptible mice, but not in the C57BL/6J mice, at 4 months. Interestingly, neither *Cd40* (the gene encoding Cd40) nor *Cd40l* (the gene encoding the Cd40 ligand) were upregulated in the salivary glands of C57BL/6.NOD-*Aec1Aec2* mice, while C57BL/6J mice actually showed minimal, but not significant, *CD40lg* upregulation (Figure 4).

#### 2.1.4. The Appearance of Infiltrating Immune Cells in the Salivary Glands Is Regulated by Chemotactic Factors

MZB cells express multiple chemokine receptors, especially Cxcr4, Cxcr5 and Ccr7. As presented in Figure 5, the genes encoding these chemokine receptors are highly upregulated in the salivary glands of C57BL/6.NOD-*Aec1Aec2* mice at 4 months of age, along with the genes encoding their chemokine ligands Cxcl12, Cxcl13, Ccl19 and Ccl21. The results also indicate upregulation of the *Ccr6* gene expression, but apparently not of its ligand *Ccl20* gene. As expected, C57BL/6J mice did not show any evidence of upregulated expressions for this same set of chemokine receptors or their respective chemokine ligands. Interestingly, while chemokine genes *Ccr1*, *Ccr2*, *Ccr3*, *Ccr8*, *Ccr9* and *Ccr10* showed no upregulation in the salivary glands of C57BL/6.NOD-*Aec1Aec2* mice, suggesting limited or no involvement of neutrophils at this timepoint, microarray data indicated upregulated expressions of *Ccr1*, *Ccr2* and *Ccr9* in the salivary glands of C57BL/6J mice at 8 weeks of age. 

#### 2.1.5. A Major Function of the Notch2 Signaling Pathway Is to Activate Mitogen-Activated Protein Kinase 14 (p38Mapk14α)

Although each component within this pathway is essential in dictating the specific characteristics and functions of the final immune profile, there are four interesting critical factors of regulation that help determine MZB cell differentiation and activation. These are Taok3 (Tao kinase 3), Adam10 (a disintegrin and metallopeptidase domain), nicastrin (Ncstn) and Bruton’s tyrosine kinase (Btk) [39,40,41,42,43]. Taok3 is a factor expressed in virtually all tissues, but functionally is involved in committing T1 B cells to the MZB cell fate by mediating surface expression of Adam10 via BCR and Notch2 ligand pathways. Adam10 is expressed on mature MZB cells, but not on follicular B cells. Nicastrin, a member of the γ-secretase complex, plays an important role in the transitioning of T2 B cells towards MZB cells by glycosylating γ-secretase. In contrast, Btk blocks differentiation of T2 B cells into MZB cells, instead promoting the differentiation of T2 B cells into follicular B cells. As presented in Figure 6, the genes encoding Taok3, Adam10 and nicastrin are upregulated in C57BL/6.NOD-*Aec1Aec2* mice at this early age, while the gene encoding Btk is not. Interestingly, *Taok3* gene expression reveals a different profile from *Adam10* and *Ncstn*. This profile is strongly conducive for the activation of p38Mapk14α. In control C57BL/6J mice, only *Taok3* is strongly upregulated, most likely involved in other cellular events.

## 3. Discussion

Over the past two decades, we have studied numerous murine models of Sjögren’s syndrome to identify the molecular and cellular factors involved in development and onset of the pathology associated with destruction and loss of the excretory function observed in the salivary and lacrimal glands of this autoimmune disease. While it was clear that a destructive T cell-mediated autoimmune process was involved in the onset of the clinical disease, we were surprised to find that SS was fully dependent on the presence of B cells [17]. More recently, it has become clear that the B cell population of critical importance in establishing an SS-like phenotype is the MZB cell per se. This discovery raises the question of what molecular bioprocesses elicited by MZB cells are relevant to the onset of SS autoimmunity. 

The overall characteristic profile described for mouse MZB cells is consistent with their broad functional ability to mount both T-independent and T-dependent antigenic immune responses, as well as regulate both innate and adaptive immunity. This unique capacity results from their inherent expressions of polyreactive BCRs, TLRs, S1P receptors and C’ receptors in conjunction with their proclivity for retention in marginal zones (MZs), rather than circulating through the blood and lymph systems. This is due in part to strong binding with integrins αLβ2 and α4β1 expressed on MZ stromal cells and the scavenger receptor Marco (macrophage receptor with collagenous structure) expressed on MZ macrophages [44,45,46,47]. Retention of MZB cells in mouse MZs permit their BCRs direct access to blood-borne antigens presented by dendritic cells (DCs) and granulocytes (GCs). In addition, antigens decorated with complement opsonins bind to the C’ receptors, while an array of antigens can bind directly to TLRs and S1P receptors. In turn, antigen-activated MZB cells possess the capability to present antigens to T lymphocytes or, with costimulatory signals, differentiate into IgM-secreting plasmablasts that can undergo further class switching in the correct milieu. Lastly, MZB cells express several chemokine receptors, in particular, Ccr5, whose ligand is Cxcl13. At high concentrations, Cxcl13 is capable of dislodging MZB cells from MZs, permitting migration to sites of injury [48]. The current data indicate a marked upregulation of the *Cxcl13* gene in the salivary glands of our SS-susceptible mice at 4 months of age correlating with the appearance of infiltrating B cells.

As stated, MZB cells are important players in innate responses and capable of inducing both T cell-dependent and T cell-independent responses. In T cell-independent responses, MZB cell activation can occur in MZs through presentation of antigens to their BCRs by reticular cells, neutrophils and dendritic cells or to their TLRs by macrophages. Whereas neutrophils and dendritic cells release both Baff and April (which in turn activates TACI), neutrophils also secrete IL-21, while dendritic cells release chemokine ligands Cxcl9, Cxcl10 and Cxcl11 (which bind Cxcr3). IL-21 is involved in class–switch recombination and somatic hypermutation, possibly leading to production of IgA and IgG. No evidence was seen for upregulation of *Il21* gene expression out to 20 weeks of age (data not presented) and may suggest an absence of B cell class switching in the salivary gland of our SS-susceptible mice. In contrast, *Cxcr4* and its ligand *Cxcl2,* as well as *Ccr7* and its ligands *Ccl19* and *Ccl21*, are all upregulated, which most likely suggests the initial recruitment and activation of a T cell-mediated response as indicated by the appearance of CD3+ T cells in lymphocytic foci of the salivary gland while B cells are still present in the glands (Figure 1). 

In contrast to the cellular processes activated in murine MZB cells responding against T cell-independent antigens, murine MZB cells responding towards T cell-dependent antigens can utilize several uniquely different molecular processes. In the first situation, Dcir2+ MZ dendritic cells capture and present antigens to both MZB cells and CD4+ T cells. This event promotes the T cell population to differentiate into Th2 cells that secrete Cd40 ligand and IL4, providing an environment for the MZB cell to transition to an antigen-presenting cell and differentiate to an IgG_1_-producing plasmablasts. However, neither *Cd40* nor *Cd40* ligand exhibited upregulated expression within the salivary gland. In the second situation, MZB cells that are lipid-reactive can present various glycolipids via Cd1d molecules to invariant natural killer T (iNKT) cells. Production of IL4-, Cd40 ligand-, and Ifnγ-activated iNKT cells, in turn, can induce differentiation of the MZB cells to IgM/IgG plasmablasts. In the third scenario, MZB cells that capture complement-opsonized antigens downregulate their S1P receptors, which allows the Cxcr5+ MZB cells to be recruited by chemokine Cxcl13 to splenic and/or nodal follicles where they deposit captured antigens to follicular DCs. Although it was reported that this migration of the MZB cells is dependent on LSC (coded by *Arhgef1*), more recent studies have indicated that Arhgef6 can also perform this function [49]. Our profiling indicates the *Arhgef6* gene and not the *Arhgef1* gene is upregulated in the salivary glands of our SS mouse model (unpublished data), further defining the molecular events more precisely. Nevertheless, in this manner, follicular T and B cells can be activated resulting in a systemic destructive T cell-mediated immune response.

## 4. Materials and Methods

### 4.1. Animals

Microarray studies were carried out using C57BL/6.NOD-*Aec1Aec2* mice as the experimental model for primary SS-like disease and C57BL/6J mice as the comparative control strain. C57BL/6.NOD-*Aec1Aec2* mice exhibit: (a) aberrant apoptosis of glandular cells by 2 months of age, (b) transient macrophage infiltrations between 2–3 months of age, (c) auto-antibody production by 3–4 months of age, (d) lymphocytic infiltration and formation of lymphocytic foci in the salivary and lacrimal glands starting between 3–4 months of age and (e) decreased glandular secretion by 4–5 months of age [16,33,34,35]. These mice, however, do not routinely progress to a lymphomagenesis stage. Mice that develop signs of dry eye disease are treated with eye salve. C57BL/6J control mice may rarely show low levels of salivary gland infiltration, but do not develop manifestations of an SS-like disease. All animals were maintained on a 12-h light/dark schedule and provided food and acidified water ad libitum. Mice were euthanized at the times indicated by cervical dislocation after deep anesthetization. This procedure does not affect preparation of tissues or RNA.

### 4.2. Histology

Salivary glands surgically removed from each mouse at time of euthanasia were placed in 10% phosphate-buffered formalin for 24 h, embedded in paraffin and sectioned at 5 µm thickness. Following deparaffination and dehydration, each section was treated with blocking solution containing donkey serum. Sections were then stained with purified rat anti-mouse CD45R (Clone 30-F11, BD Pharmingen, San Jose, CA, USA) diluted 1:25 and goat polyclonal IgG anti-mouse CD3ε (Clone M-20, Santa Cruz Biotechnology, Santa Cruz, CA, USA) diluted 1:50 in an antibody diluent (Dako, Carpinteria, CA, USA) for 1 h at 25 °C. The slides were then washed with PBS followed by a 1 h incubation with Alexa Fluor 488 donkey anti-goat IgG (H+L) and Alexa Fluor 594 donkey anti-rat IgG (H+L) (Life Technologies, Grand Island, NY, USA). After a thorough wash with PBS, the slides were treated with a Vectashield DAPI-mounting medium (Vector Laboratory, Burlingame, CA, USA) and visualized microscopically at 200×.

### 4.3. RNA Preparation

Procedures for the isolation, preparation and quality testing of RNA samples were published in detail elsewhere [29,36]. In brief, salivary glands, free of lymph nodes, were excised in parallel from C57BL/6.NOD-*Aec1Aec2* and C57BL/6J mice at each of the five age points indicated (*n* = 5 mice per age group), snap-frozen in liquid nitrogen and stored at −80 °C until all samples were obtained. Using one lobe of each salivary gland comprised of the submandibular, sublingual and parotid gland, total RNA from each age group and both strains were isolated concurrently using a RNeasy Mini-Kit (Qiagen, Valencia, CA, USA), as per the manufacturer’s protocol. 

### 4.4. Generation of Transcriptome Data

Each salivary gland RNA sample was hybridized onto an Affymetrix 3′ Expression Array GeneChip Mouse Genome 430 2.0 array and annotated (build 32; 06.09.2011), while verification of the RNA results was carried out on randomly selected genes using real-time polymerase chain reaction (rt-PCR) analyses. Microarray data are deposited with Gene Expression Omnibus, accession numbers GSE15640 and GSE36378. A recent extensive and fully detailed analysis of our transcriptome data can be seen in our previous publication [31]. For the current manuscript, we present some abbreviated transcriptome datasets, e.g., 12-, 16- and 20-week-old SS-susceptible (SS^S^) C57BL/6.NOD-*Aec1Aec2* mice or 12- and 16-week-old SS-non-susceptible (SS^NS^) control C57BL/6J mice, as these are timeframes for early lymphocytic foci development in the salivary glands of C57BL/6.NOD-*Aec1Aec2* mice.

### 4.5. Gene expression Data Analysis

Analysis of gene expression has been detailed elsewhere [29]. In brief, microarray data were normalized using the robust multiarray average (GCRMA) algorithm and the Linear Models for Microarray Analysis (LIMMA) package (http://www.r-project.org) for differential expression analyses. For the current study, the fdr method of Benjamini and Hochberg [50] was used to adjust the *p*-values for multiple testing as a means to control the false discovery rate. The original data represent 5 equally spaced timepoints; thus, multiple models were used to identify temporal patterns of gene expression, i.e., the linear fit (degree = 1), quadratic fit (degree = 2), cubic fit (degree = 3) and quartic fit (degree = 4) regression models. B-statistics were calculated for each gene providing odds that a gene shows either positive or negative trends over time. In addition, the temporal relative expression profiles together with the comparison of SS^S^ and SS^N^ individual gene values relative to the value at the 4 week timepoint were considered. Data presented show the differential expression of an individual gene’s value at the 8, 12, 16 or 20 weeks timepoints relative to that gene’s value at the 4 weeks timepoint, the time when the salivary glands are mature and pre-diseased. Comparison of gene values between C57BL/6J and C57BL/6.NOD-*Aec1Aec2* always involved the same aptamers. Relative differential gene expressions were verified using rt-PCR to quantitate relative temporal RNA levels.

## 5. Conclusions

The transcriptome data profiles presented in the current study indicate the presence of an MZB cell population in the salivary glands of our SS-susceptible C57BL/6.NOD-*Aec1Aec2* mouse model at 4 months (16 weeks) of age, the time when autoantibodies are detected in sera and lymphocytic foci rapidly form in salivary and lacrimal glands [14,16]. A Notch2 signal transduction profile in the salivary glands revealed an MZB cell population undergoing both ontogenesis and activation, with the latter persisting in the long term. Considering the importance of MZB cells in the development and onset of SS in various mouse models, it is not unexpected to see a transcriptomic profile defining MZB cells in the salivary glands. One advantage of using temporal global transcriptome analyses is the possibility to define molecular bioprocesses central to development of an autoimmune pathology observed both histologically and pathologically, as these two parameters are in constant flux. In conjunction with our previously published studies indicating absolute requirements for Ig BCR [17], C’3 [51,52], IL4 [53] and IFNγ/IFNR [54] in the development of the SS disease in our SS^S^ mice, our transcriptome analyses presented here suggest the third scenario described above is the central bioprocess occurring in the salivary glands at around 4 months of age. This, therefore, represents a potential focal point for intervention therapy just prior to or at the start of activation of the T cell-mediated cytotoxic response. Nevertheless, it is important to note that this transcriptome analysis strictly reflects the ongoing bioprocesses in the salivary gland and fails to address any active bioprocesses occurring in the follicles of the draining lymph nodes or spleen [55], sites that are most likely critical in development of the adaptive T cell-mediated response observed later in the overt clinical disease.

## Figures and Tables

**Figure 1 ijms-22-01919-f001:**
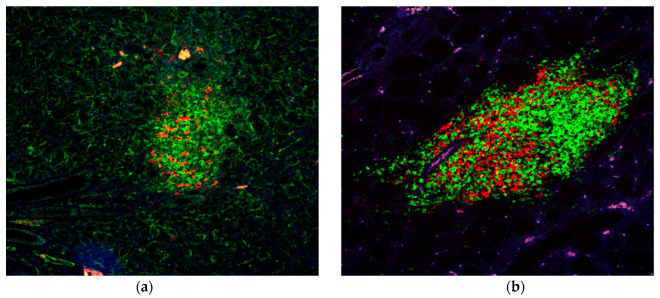
Histological photomicrographs depicting lymphocytic infiltrations (lymphocytic foci (LF)) present within the salivary glands of both C57BL/6J (**a**) and C57BL/6.NOD-*Aec1Aec2* mice (**b**) at 7 months of age. While the infiltrations within the salivary glands of both mouse strains contain both T cells (green fluorescence) and B cells (red fluorescence), both the transcriptomes and immunological activities are very different from each other. During the early disease stages, LF of our SS^S^ mice contain higher numbers of B cells; however, at later stages, the T lymphocyte population(s) tend to overtake the number of B lymphocytes. In addition, the LF of the SS^S^ mice are strongly periductal.

**Figure 2 ijms-22-01919-f002:**
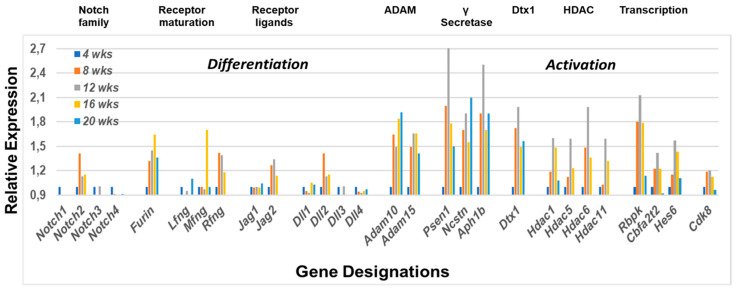
Prolonged upregulated activation of the Notch2 signal transduction pathway in the salivary glands prior to onset of the adaptive response. Notch2 is a transmembrane protein receptor whose extracellular portion is post-translationally modified in the Golgi bodies by furin and fringe molecules (i.e., Lfng, Mfng or Rfng) prior to insertion into the cellular membrane, enhancing subsequent MZB cell functions. On activation by a Notch2 ligand (i.e., Jagged or Delta), the Notch2 molecule undergoes sequential cleavage first by Adam10 and then by γ-secretase, thereby releasing the cytoplasmic protein Dtx (an E3 ubiquitin ligase), as well as the Notch intracellular domain (NICD) that gets transported to the nucleus. In the nucleus, NICD displaces corepressors, including HDAC (histone deacetylase complex), permitting direct interaction with Rbpk to activate Hes, thus driving Notch2-regulated gene transcription. Downregulation of the Notch2 pathway occurs through phosphorylation of NCID by Cdk8 that initiates polyubiquitination and proteasome degradation. In this model, however, while the Notch2 pathway remains activated, Cdk8 exhibits, at best, minimum upregulated expression, suggesting the MZB cells may persist and over time differentiate from their effector function to their APC function for invoking the adaptive response. Gene symbols and gene descriptions are: **Notch** (notch gene homolog (*Drosophila*)); ***Furin*** (paired basic amino acid-cleaving enzyme); ***Lfng*** (LFNG O-fucosylpeptide 3-beta-acetylglucosaminyltransferase); ***Mfgn*** (MFNG O-fucosylpeptide 3-beta-n-acetylglucosaminyltransferase); ***Rfng*** (RFNG O-fucosylpeptide 3-beta-acetylglucosaminyltransferase); ***Jag1*** (jagged 1); ***Jag2*** (jagged 2); ***Dll1*** (delta-canonical Notch ligand/delta-like 1 homolog); ***Dll2*** (delta-canonical Notch ligand/delta-like 2 homolog); ***Dll3*** (delta-canonical Notch ligand/delta-like 3 homolog); ***Dll4*** (delta-canonical Notch ligand/delta-like 4 homolog); ***Adam10*** (a disintegrin and metallopeptidase domain 10); ***Adam15*** (a disintegrin and metallopeptidase domain 15); ***Psen1*** (Presenilin 1); **Ncstn** (nicastrin); **Aph1b** (aph-1 homolog B, gamma-secretase); ***Dtx4*** (deltex 4 homolog (*Drosophila*)); ***Hdac1*** (histone deacetylase 1); ***Hdac5*** (histone deacetylase 5); ***Hdac6*** (histone deacetylase 6); ***Hdac11*** (histone deacetylase 11); ***Rbpk*** (recombination signal binding protein for the IgK J region); ***Cba2t2*** (partner transcriptional co-repressor 2); ***Hes6*** (hairy and enhancer of split 6 (*Drosophila*)); ***Cdk8*** (cyclin dependent kinase 8).

**Figure 3 ijms-22-01919-f003:**
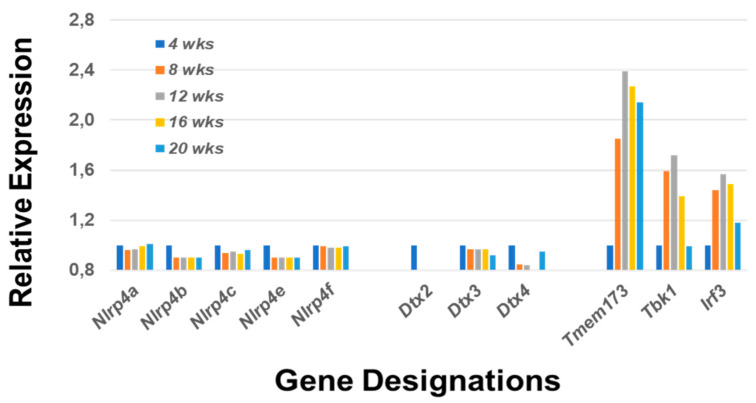
Prolonged upregulated activation of the type 1 IFN response correlates with downregulated Dtx expression. Regulation of type 1 IFN is, in part, dependent on activation of deltex family protein Dtx4 (deltex E3 ubiquitin ligase 4) via Nlrp4 (NLR family, pyrin domain-containing 4). The Nlrp4 complex regulates IFN synthesis by targeting Tbk1 (Tank-binding kinase 1) for degradation by Dtx4. Tbk1 is a critical component in the activation pathway involving *Tmem173* (Sting), *Tbk1* and *Irf3* (interferon regulatory factor *3*). As neither *Dtx4* nor any one of the possible *Nlrp4* genes expressed in mice (i.e., *Nlrp4a, b, c, e* and *f*) exhibit temporal upregulated expression, it is predicted that a strong prolonged IFN response would occur in the SS-susceptible C57BL/6.NOD-*Aec1Aec2* mice during the disease state giving rise to the IFN signature characteristic of SS.

**Figure 4 ijms-22-01919-f004:**
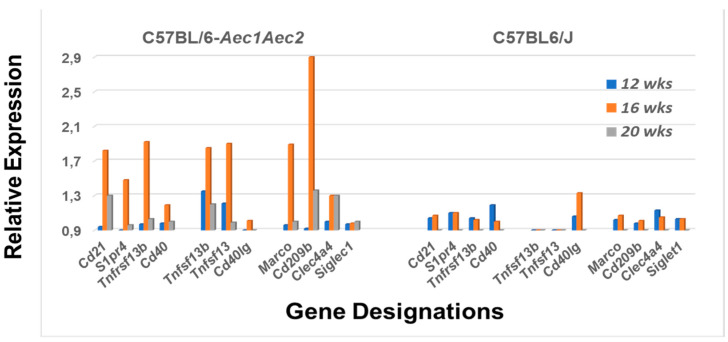
Differential expression profiles for genes encoding various receptors associated with the MZB cell function. Comparative transcriptome data showing the temporal expression of MZB cell-associated receptor genes (*Cd21*, *S1pr4*, *Tnfrsf13b* and *Cd40*), receptor ligands (*Tnfsf13b*, *Tnfsf13* and *Cd40lg*) and macrophage receptors (*Marco*, *Cd209b*, *Clec4a4* and *Siglec1*) in salivary glands of SS^S^ C57BL/6.NOD-*Aec1Aec2* mice (**left**) versus SS^NS^ C57BL/6J mice (**right**).

**Figure 5 ijms-22-01919-f005:**
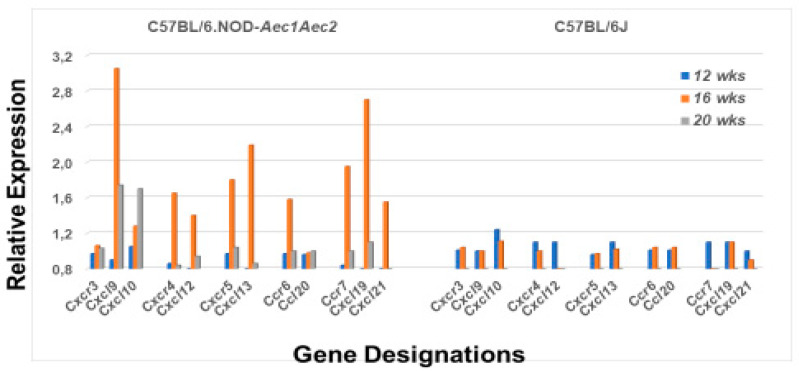
Differential expression profiles for genes encoding chemokine and chemokine receptors associated with MZB cell activity. Comparative transcriptome data showing the temporal expression of MZB cell-associated chemokine receptor (chemokine (CXC motif) receptor) genes (*Cxcr3*, *Cxcr4*, *Cxcr5*, *Cxcr6* and *Cxcr7*) and their respective chemokine ligand (chemokine (CXC motif) ligand) genes (*Cxcl9, Cxcl10, Cxcl12, Cxcl13, Ccl20, Cxcl19* and *Cxcl21*) in salivary glands of SS^S^ C57BL/6.NOD-*Aec1Aec2* mice (**left**) versus SS^NS^ C57BL/6J mice (**right**).

**Figure 6 ijms-22-01919-f006:**
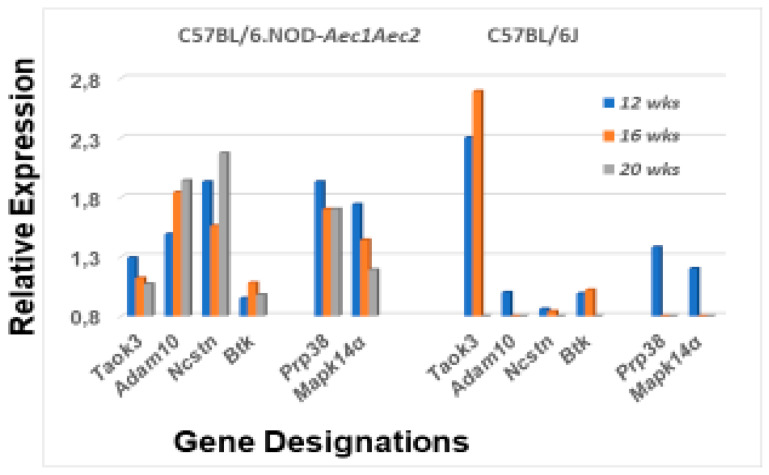
Differential expression profiles for genes encoding various factors involved in the regulation of MZB cell signal transductions. Comparative transcriptome data showing the temporal expression of genes encoding factors critical in controlling the Notch2 signal transduction pathway of MZB cells, i.e., *Taok3* (TAO kinase 3), *Adam10*, *Ncstn* (nicastrin), *Btk* (Bruton’s tyrosine kinase), *Prp38* (proline rich protein 38) and *Mapk14a* (mitogen-activated protein kinase 14a) in salivary glands of SS^S^ C57BL/6.NOD-*Aec1Aec2* mice (**left**) versus SS^NS^ C57BL/6J mice (**right**).

## Data Availability

Microarray data are publicly available and deposited with The Gene Expression Omnibus, Accession Numbers GSE15640 and GSE36378.

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
