# Peer review of "Early Covert Appearance of Marginal Zone B Cells in Salivary Glands of Sjögren′s Syndrome-Susceptible Mice: Initiators of Subsequent Overt Clinical Disease"

_ijms, 2021, doi:10.3390/ijms22041919_

Round 1
Reviewer 1 Report
Minor Comments:
- Line 38: either B cell or B lymphocyte; please choose one across the MS.
- Section 2.1.1, lines 101-106: please re-phrase the section starting with the aim and mention by which approach you measure the LF.
- Lines 187-192: please state the approach with concise methodology as readers understand the CD markers and targets, for instance in Fig 1.
- Line 111-113: the sentence is vague and please cite a reference to the signaling line if you have extracted from a reference unless you have extrapolated from the RNA-seq data which is not acceptable to define a signaling pathway hypothetically based on mere expression profile.
- Line 115: Please set across the MS all unanimous abbreviations such as NF-κB etc.
- The figure’s captions and axes are blurry and with low pixels.
- Please make an abbreviation list of genes that should be added to the supplementary.
- Please re-phrase the Figures’ legends as they must include the approach and unit for relative gene expression. If it was quantitated microarray data that should be expressed as RPMK or etc.
Author Response
Reviewer 1
I wish to thank you for taking time to critique this manuscript. In response to the critique I have made multiple small changes including replacing Figure 1 and the accompanying text. Below I discuss my changes to the overall manuscript based on your specific points.
- Line 38: either B cell or B lymphocyte; please choose one across the MS. The term B cell has replaced the term B lymphocyte throughout the manuscript, even though the two terms are interchangeable. Normally I use Lymphocyte generally for B and T cells, but when talking about MZB lymphocytes, for example, they are almost universally referred to as MZB cells. Thus, the use of B cells throughout.
- Section 2.1.1, lines 101-106: please re-phrase the section starting with the aim and mention by which approach you measure the LF. In this manuscript, the purpose of presenting Figure 1 is to visually show the immune attack against the salivary gland, not to enumerate the levels, numbers and/or size of LF often performed to indicate SS severity. Nevertheless, as stated below, Figure 1 has been replaced.
- Lines 187-192: please state the approach with concise methodology as readers understand the CD markers and targets, for instance in Fig 1. In response to this comment and that of Reviewer 2, Figure 1 has been replaced with a new histological picture, plus text rewritten that slightly changes the overall aim and presentation. (lines 101-107).
- Line 111-113: the sentence is vague and please cite a reference to the signaling line if you have extracted from a reference unless you have extrapolated from the RNA-seq data which is not acceptable to define a signaling pathway hypothetically based on mere expression profile. First, this is not RNA-seq data. Second, the Notch pathway is well-defined in multiple pathway databases such as Kegg, Panther, on-line figures, numerous publications, etc. Using these multiple references to provide the basic notch signal transduction pathway, the gene set making up this well-defined pathway was examined for similar temporal upregulations within the pathway. The results, in turn, permit reconstruction of the hypothetical targeted pathway and its downstream activations. Functionally, the data are further compared to published papers, i.e., #23 and #24, to construct the hypothetical pathway in SS.
- Line 115: Please set across the MS all unanimous abbreviations such as NF-κB etc. · Please make an abbreviation list of genes that should be added to the supplementary. I have no idea what is meant by unanimous abbreviations. Each gene mentioned in the manuscript is first stated by an official symbol identifying the official name, followed by its gene title, e.g., Adam10 (A disintegrin and metallopeptidase domain10) or Cxcl# (Chemokine (cxcl motif#) ligand).
- The figure’s captions and axes are blurry and with low pixels. In addition to Figure 1, both Figure 2 and Figure 3 have also been replaced with new versions. but not content.
- Please re-phrase the Figures’ legends as they must include the approach and unit for relative gene expression. If it was quantitated microarray data that should be expressed as RPMK or etc. The data presented in Figures 2, 3, 4, 5, and 6 are all generated by microarray carried out at the same time, thus under identical conditions. The approach and gene expressions are identical, as is the aim to profile MZB cell activated bioprocesses in early-stage SS. This is presented in the Materials & Methods section. Additional information is now provided with respect to how the relative expressions were generated and subsequently treated statistically (lines 370-406).
Reviewer 2 Report
In the article entitled “Short-term covert appearance of marginal zone B cells (MZB) in salivary glands of Sjogren’s syndrome –susceptible mice: initiators of subsequent overt clinical disease” the authors, using transcriptomic analyses of C57BL/6.NOD-Aec1Aec2 mice salivary gland tissue, in different time points during the salivary gland pathology evolution of, indicated the appearance of Notch2-positive cells (MZB) within early stages of sieladenitis and identify cellular bioprocesses relative to the recruitment of these cells in the lesions.
This is an interesting approach for the delineation of the pathogenesis in a mouse disease model. The results of this study rise interest, since the lymphoma observed in patients with SS is mainly of MZB origin related to mucosa. However the composition of the lesion in the salivary gland of SS patients seems to have a quite different architecture and cell composition (J Autoimmunity 34 (2010) 400e407 ).
The data is well presented, however
- How many animals were studied?
- Histopathology is a well-known procedure and is not necessary to be extensively presented in the text
- The centrally located T-cells with the surrounding B-cells, in the early lesion, is not characteristic of a germinal center like lymphocytic foci.
- Some typographical errors should be corrected.
In the article entitled “Short-term covert appearance of marginal zone B cells (MZB) in salivary glands of Sjogren’s syndrome –susceptible mice: initiators of subsequent overt clinical disease” the authors, using transcriptomic analyses of C57BL/6.NOD-Aec1Aec2 mice salivary gland tissue, in different time points during the salivary gland pathology evolution of, indicated the appearance of Notch2-positive cells (MZB) within early stages of sieladenitis and identify cellular bioprocesses relative to the recruitment of these cells in the lesions.
This is an interesting approach for the delineation of the pathogenesis in a mouse disease model. The results of this study rise interest, since the lymphoma observed in patients with SS is mainly of MZB origin related to mucosa. However the composition of the lesion in the salivary gland of SS patients seems to have a quite different architecture and cell composition (J Autoimmunity 34 (2010) 400e407 ).
The data is well presented, however
- How many animals were studied?
- Histopathology is a well-known procedure and is not necessary to be extensively presented in the text
- The centrally located T-cells with the surrounding B-cells, in the early lesion, is not characteristic of a germinal center like lymphocytic foci.
- Some typographical errors should be corrected.
-
Author Response
Reviewer 2
I wish to thank you for taking time to critique this manuscript and provide constructive remarks. Based on the comments, I have replaced Figure 1 and made appropriate changes to the text. Below I have addressed the stated points.
- How many animals were studied? Although originally stated as n=5, the overall number of animals studied per time point is now better defined and stated (line 368)
- Histopathology is a well-known procedure and is not necessary to be extensively presented in the text. The procedure for histological staining has been rewritten in a more concise manner (lines 352-364)
- The centrally located T-cells with the surrounding B-cells, in the early lesion, is not characteristic of a germinal center like lymphocytic foci. In response to this comment, Figure 1 has been completely replaced with new histological pictures, and text rewritten to slightly change the presentation. (lines 101-107).
- Some typographical errors should be corrected. Unfortunately, the final text continued to have accent marks within several words. due to the Spanish spell check on the computer. These have been corrected, as well as a few misspelled words.